# Elastoplastic Analysis of Frame Structures Using Radial Point Interpolation Meshless Methods



Jorge Belinha [1,*], Miguel Aires [2] and Daniel E.S. Rodrigues [1,3,*]

1    Department of Mechanical Engineering, School of Engineering, Polytechnic of Porto, Rua Dr. António Bernardino de Almeida, n.431, 4200-072 Porto, Portugal
2    Faculty of Engineering, University of Porto, Rua Dr. Roberto Frias, 4200-465 Porto, Portugal; up201404912@fe.up.pt
3    Department of Mechanical Engineering, University of Aveiro, Campus de Santiago, 3810-193 Aveiro, Portugal
*    Correspondence: job@isep.ipp.pt (J.B.); der@isep.ipp.pt or desr@ua.pt (D.E.S.R.)

**Abstract:** The need to design structures and structural elements that are more efficient in terms of performance is a key aspect of engineering. For a given material to be used at its maximum capacity, considering non-linear characteristics is mandatory. The non-linear regime is a subject of extreme interest for this reason and is an area with intense research activity. In this work, advanced discretization techniques (i.e., meshless methods) are applied in the elastoplastic analysis of 2D and 3D structural elements. The literature shows that meshless methods are capable of producing more accurate and smoother strain and stress fields, which are the variable fields required in the non-linear models describing elastoplasticity. Thus, in this study, the Radial Point Interpolation Method (RPIM) and the Natural Neighbor Radial Point Interpolation Method (NNRPIM) are combined with a non-linear iterative algorithm, fully developed by the authors, with the objective of analyzing for the first time the elastoplastic behavior of a two-bay asymmetric frame and bowstring bridge considering 2D and 3D analysis. The accuracy and robustness of the RPIM and the NNRPIM are shown in the end, comparing the obtained results with FEM solutions and the available literature.

**Keywords:** meshless methods; radial point interpolation method; elastoplasticity; frame structures

## 1. Introduction

In Computational Mechanics, the Finite Element Method (FEM) is a standard numerical tool capable of analyzing several problems. However, when some of those problems involve mesh distortions, the FEM may be not the most suitable method [1]. Meshless methods are an alternative to standard methods like the FEM. In these methods, the domain is discretized by a random set of nodes rather than an element mesh. Then, the influence-domain concept is applied instead of an element concept, so the nodes can be distributed in a random fashion [2,3]. These and other characteristics make meshless methods ideal for computing problems involving complex and transitory geometries [4] and non-linear processes involving the iterative use of strain and stress fields [5]. In this work, such problems are studied: the elastoplastic behavior of frames and structures.

The mechanical behavior of structures is often analyzed using a linear analysis. However, the existence of non-linear phenomena applied to said structures during their working conditions is a reality, and they need to be accounted for. Non-linear behavior can bring certain economic and safety benefits to structural projects, given that after the material surpasses its yield limit, it is not necessarily true that the structure composed of said material is on its way to collapse. In reality, that same structure can still withstand loading. Accounting for elastoplastic behavior leads to a more complete project, not only in safety terms but also in economic terms, with dimensions of profiles and structural elements being inferior to the ones used in all-elastic domain design. The elastoplastic theory is a mandatory subject in the current need for thinner, lighter, and more mechanically resistant

structural elements since accounting for the elastoplastic effects is a way to reduce costs and increase the safety of structural elements.

As previously stated, this work will use meshless methods to simulate the elastoplastic behavior of several structures. Meshless methods first appeared with the Smooth Particle Hydrodynamics Method (SPH) in 1977 [6], applied to solid mechanics by Larry D. Libersky and A.G. Petschek [7] only in 1991. The SPH uses a strong formulation of the physical problem in order to solve it. On the other hand, the Element-Free Galerkin Method (EFGM) [1] is the first to depend on a global weak form. Some other mesh-free methods appeared at the same time, like the Reproducing Kernel Particle Method (RKPM) [8] or the Meshless Local Petrov–Galerkin method [9]. The Natural Element Method (NEM) [10] was the first proposed method to solve the lack of the delta Kronecker property. Over the course of the next ten years, other methods appeared such as the Method of Finite Spheres (MFS) [11], the Point Interpolation Method (PIM) [12], the Point Assembly Method (PAM) [13], the Meshless Finite Element Method (MFEM) [14], the Natural Neighbor Radial Point Interpolation Method (NNRPIM) [15] and the Radial Point Interpolation Method (RPIM) [16]. The Radial Point Interpolation Method (RPIM) [16] and The Natural Neighbor Radial Point Interpolation Method (NNRPIM) [5] are used in this work.

Meshless methods have been applied in some elastoplastic simulations of structures. Zhou et al. [17] used a centroid-enriched edge-based/face-based smoothed RPIM (CE-ES-RPIM and CE-FS-RPIM) to analyze elastoplastic benchmark problems in solid mechanics; the Meshless Local Petrov–Galerkin method was applied in the thermo-elastoplastic analysis of thick functionally graded (FG) plates by Vaghefi et al. [18]; the same author studied, for the first time, three-dimensional (3D) thermo-elastoplastic bending analysis of functionally graded sandwich plates subjected to combined thermal and mechanical loads, using a local radial point interpolation method (LRPIM) [19]; the RRKPM was used in the elastoplastic analysis of functionally graded materials by Liu et al. [20]; the same meshless method was applied in the elastoplastic analyses of several solid mechanics problems by Gao et al. [21]; a novel EFGM based on the improved complex variable moving least-squares (ICVMLS) approximation (called improved complex variable element-free Galerkin (ICVEFG)), was applied to two-dimensional large deformation elastoplasticity problems [22]; the EFGM was also used in finite deformation elasto-plastic modeling [23].

In the next section, the meshless methods used throughout this work are presented: the RPIM, a simple meshless method resembling the FEM from a programming point of view, and its improved version with higher nodal connectivity, the NNRPIM. Concepts such as influence-domain and nodal connectivity are explained, including the integration scheme and the interpolation functions used in both methods. Moreover, for the NNRPIM, the notion of influence-cells and how they are constructed using mathematical concepts such as Voronoï diagrams is tackled. Then, the elastoplastic model implemented within the RPIM and the NNRPIM codes is detailed. In the end, several numerical examples are presented for a two-bay asymmetric frame and a bowstring bridge considering 2D and 3D analysis. This marks the first time in the literature that these structures are analyzed using meshless methods in a non-linear material regime. The results achieved by the two meshless methods shed light on the accuracy and robustness of these discretization techniques as a strong alternative to the traditional FEM.

## 2. Radial Point Interpolation Meshless Methods

The standard meshless procedure starts with the discretization of the problem domain into a nodal mesh, which can arbitrarily be created (Figure 1). A regular or irregular nodal distribution can be assigned to the domain. A higher nodal density generally leads to more accurate results and, also in a general case scenario, irregular distributions have a lower accuracy when compared to regular distributions.

The second step in a meshless algorithm is the construction of the integration mesh. The NNRPIM uses a nodal-dependent integration mesh to integrate the integro-differential equations from the Galerkin Weak form. On the other hand, the RPIM uses the Gauss-

Legendre quadrature and integrates the discrete system of equations using a background mesh—just like the FEM.

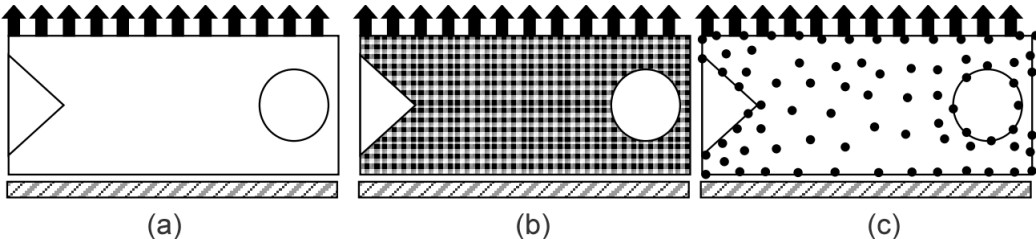

**Figure 1.** (**a**) Solid Domain. (**b**) Regular discretization. (**c**) Irregular discretization.

After that, the nodal connectivity is imposed using the influence-domain concept: for a certain interest point $\mathbf{x}_I$, the domain of influence of such a point will be an area or volume centered at said point. Thus, the nodes inside these areas or volumes will be used to interpolate the field variables at the interest point, and the overlap of the domains will ensure nodal connectivity. The procedure is similar in both RPIM and NNPRIM, although the shape of the influence-domains is different for both methods. Figure 2a shows some influence-domains with different sizes and shapes. According to [16], influence-domains with a fixed number of nodes (between $n = 9$ and $n = 16$) allow the construction of shape-functions with the same level of complexity.

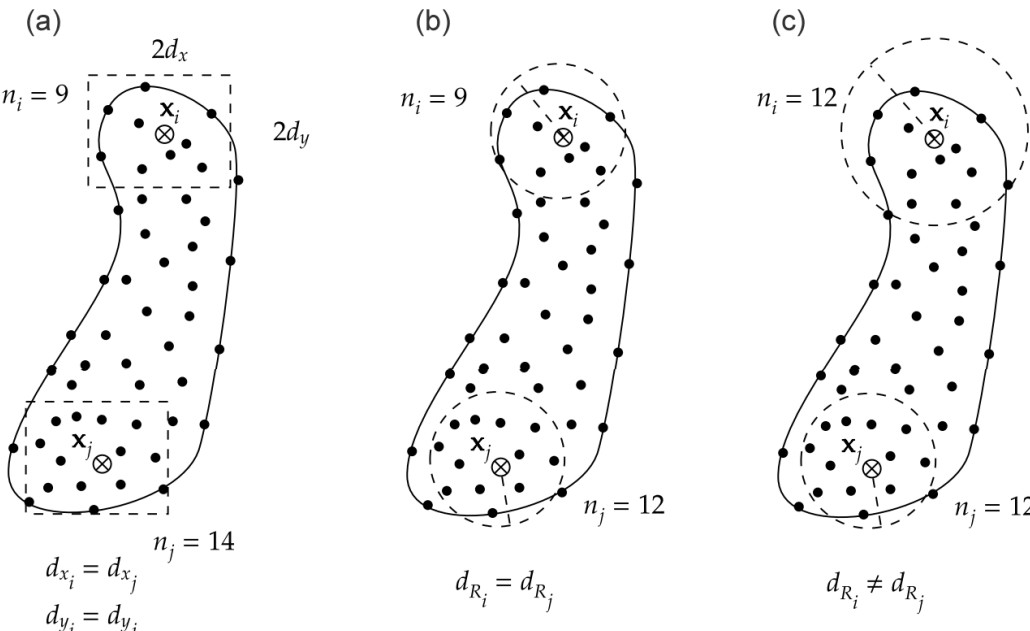

**Figure 2.** (**a**) Fixed rectangular influence-domains. (**b**) Fixed circular influence-domain. (**c**) Flexible circular shaped influence-domains.

As hinted before, the NNRPIM's influence-domains have a slightly different shape when compared with the ones used in the RPIM formulation. In fact, the domains of influence in the NNRPIM are called influence-cells because they are composed of Voronoï cells. In the NNRPIM formulation, the construction of the nodal mesh directly leads to the construction of a Voronoï diagram [24] based on it. Every Voronoï cell contains one node of the nodal mesh at its center and represents the geometric region whose points are closer to this central node than any other node outside of the cell. Thus, an influence-cell of a given interest point will be composed of the area occupied by the correspondent Voronoï cell plus the area of the Voronoï cells that are adjacent to the central one—this is the definition of

natural neighbors. For mathematical completeness, Ref. [5] provides a detailed explanation of how these geometrical constructions are established.

As a consequence of the creation of the Voronoï diagram, a nodal-dependent integration mesh is constructed, as stated before. The Delaunay tessellation is the geometrical dual of the Voronoï diagram, and it is constructed by connecting the nodes whose Voronoï cells have common boundaries. Using this geometrical concept, Belinha et al. [5] then proposed an integration scheme: the Delaunay tessellation divides the Voronoï cells into subcells, and integration points are placed at the center of the sub-cell, the integration weight of that point being related with the area occupied by such integration point—Figure 3.

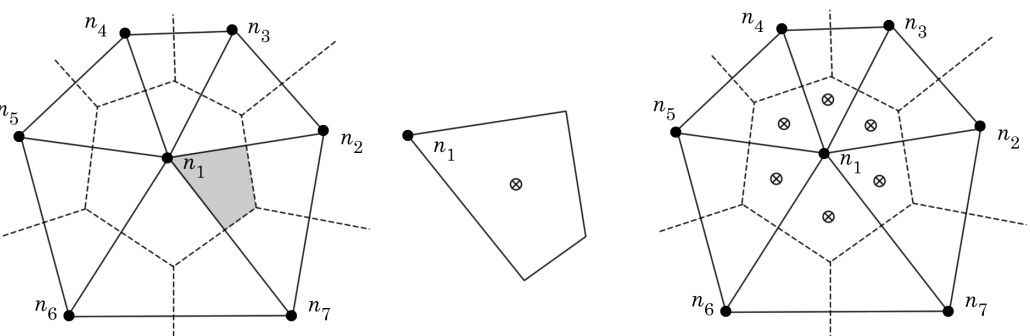

**Figure 3.** Sub-cell obtained through the overlapping of the Voronoï diagram, the Delaunay tessellation, and how quadrature points following the Gauss–Legendre integration scheme are established.

The remaining RPIM and NNRPIM formulations are similar. As stated before, the field variables are interpolated within each influence-domain. For instance, consider the displacement field $\mathbf{u}$. The displacement components $\mathbf{u}_I = (u, v, w)$ at an interest point $\mathbf{x}_I$ are computed using the nodal displacement of the nodes inside the influence-domain of said interest point:

$$\mathbf{u}(\mathbf{x}_I) = \sum_{j=1}^{n} \varphi_j(\mathbf{x}_I)\mathbf{u}(\mathbf{x}_i) \tag{1}$$

where $n$ is the number of nodes inside the influence-domain or influence-cell of the interest point $\mathbf{x}_I$, $\mathbf{u}(\mathbf{x}_i)$ is a vector containing the nodal displacements of the nodes within the influence-domain, and $\varphi_j(\mathbf{x}_I)$ is the interpolation function value at the $j$th node. Both RPIM and NNRPIM use the same interpolation functions, which are a combination of polynomial basis functions and the Multiquadratic Radial Basis Function (MQ-RBF)—proposed initially by Hardy [25]:

$$R(r_{ij}) = (r_{ij}^2 + c^2)^p \tag{2}$$

with $c$ and $p$ being shape parameters, whose optimal values are $c = 1.42$ and $p = 1.03$, according to [16,26]. Nevertheless, Belinha et al. [5] performed an optimization study for these shape parameters and concluded that $c = 0.0001$ and $p = 0.9999$ are more suitable values. These last shape parameter values were used in this work. The Euclidean distance between the point $\mathbf{x}_I$ and a node $\mathbf{x}_i$ within the same influence-domain can be calculated for 2D problems using:

$$r_{ij} = \sqrt{(\mathbf{x}_I - \mathbf{x}_i)^2 + (\mathbf{y}_I - \mathbf{y}_i)^2} \tag{3}$$

Thus, Equation (1) can be rewritten in a matrix form as:

$$\mathbf{u}(\mathbf{x}_I) = \{\mathbf{R}^T(\mathbf{x}_I), \mathbf{p}^T(\mathbf{x}_I)\}\mathbf{G}^{-1}\begin{Bmatrix} \mathbf{u}_s \\ 0 \end{Bmatrix} = \varphi(\mathbf{x}_I)\mathbf{u}_s \tag{4}$$

where $\varphi(\mathbf{x}_I)$ is the matrix containing the values of the interpolation functions of every node composing the influence-domain calculated at the interest point $\mathbf{x}_I$, and matrix $\mathbf{R}$, of

size $[n \times n]$, contains the values of the MQ-RBFs calculated by Equation (2) at the interest point $\mathbf{x}_I$:

$$\mathbf{R} = \begin{bmatrix} R(r_{11}) & R(r_{12}) & \dots & R(r_{1n}) \\ R(r_{21}) & R(r_{22}) & \dots & R(r_{2n}) \\ \vdots & \vdots & \ddots & \vdots \\ R(r_{n1}) & R(r_{n2}) & \dots & R(r_{nn}) \end{bmatrix} \tag{5}$$

Matrix $\mathbf{p}$, of size $[n \times m]$, is the matrix containing the polynomial basis (m is the monomial number, i.e., the number of terms of the considered polynomial basis):

$$\mathbf{p} = \begin{bmatrix} p_1(\mathbf{x}_1) & p_2(\mathbf{x}_1) & \dots & p_m(\mathbf{x}_1) \\ p_1(\mathbf{x}_2) & p_2(\mathbf{x}_2) & \dots & p_m(\mathbf{x}_2) \\ \vdots & \vdots & \ddots & \vdots \\ p_1(\mathbf{x}_n) & p_2(\mathbf{x}_n) & \dots & p_m(\mathbf{x}_n) \end{bmatrix} \tag{6}$$

$\mathbf{u}_s$ is the nodal displacement vector of the influence-domain of $\mathbf{x}_I$, and $\mathbf{G}$ is given as:

$$\begin{bmatrix} \mathbf{R} & \mathbf{p} \\ \mathbf{p}^T & 0 \end{bmatrix} = \mathbf{G} \tag{7}$$

For more detailed mathematical information on the interpolation functions used in the RPIM and NNPRIM, the book by Belinha et al. [5] is well documented.

With the shape functions established, the discrete system of equations can be obtained using the Galerkin Weak Form. This is a rather standard procedure, well documented in the literature [5], and therefore it will not be presented in this paper.

In comparison with the FEM, the RPIM and the NNRPIM possess higher nodal connectivity (due to the overlap rule of influence-domains), which usually leads to more accurate solutions, even for non-transitory geometry problems. The accuracy of the meshless methods used in this work will be proven in Section 4. Additionally, the interpolation functions cited here have virtually a higher order, allowing a higher continuity and reproducibility. Due to the preprocessing of the Voronoï diagram and the construction of the node-dependent integration mesh, higher computational costs are expected in the NNRPIM, this being its main disadvantage.

## 3. Elastoplastic Model

Plasticity models depend on the nature of the materials whose behavior is being simulated. In Figure 4, some elastoplastic models are observed.

In Figure 4a, we can see that the elastic part is linear, characterized by the Young Modulus $E$ and by $\alpha_1 = \tan^{-1} E$. When the yield stress, $\sigma_Y$, is reached, the stress level is maintained while the deformation, of plastic nature only, increases. In Figure 4b, we have two slopes. The first slope, characterized by $\alpha_1$, is the elastic part, while the second slope, when the yield stress is reached, characterized by $\alpha_2 = \tan^{-1} E_T$, is the plastic part, with $E_T$ being the tangent modulus. In the case of Figure 4c, the stress relation after yield is no longer linear. This work uses a linear elastic–linear plastic model similar to Figure 4b.

In this work, a small strain formulation is used to model the elastoplastic behavior of structures. In small strains, the total strain $\varepsilon$ is separated into two components [27,28]. This is called the decomposition law:

$$\varepsilon = \varepsilon^p + \varepsilon^e \tag{8}$$

where $\varepsilon^p$ is the plastic component of the total strain and $\varepsilon^e$ is the elastic component.

The yield criterion, which assesses if the material is under an elastic or plastic regime, can be generally defined as a function of the stress, $\sigma$, and the load history:

$$F(\sigma, \kappa) = f(\sigma, \kappa) - \sigma_Y(\kappa) = 0 \tag{9}$$

where $f(\sigma, \kappa)$ is a yield function that depends on the stress state, $\sigma$; a hardening parameter, $\kappa$; and the yield stress of the material, $\sigma_Y(\kappa)$, which can also be hardening-dependent. In this study, to evaluate if the stress level surpasses the yield surface, the von Mises yield criterion is used:

$$\bar{\sigma}^2 = \sigma_Y^2 = 3J_2 \tag{10}$$

where $\bar{\sigma}$ is the effective stress, and $J_2$ is the second invariant of the deviatoric stress tensor.

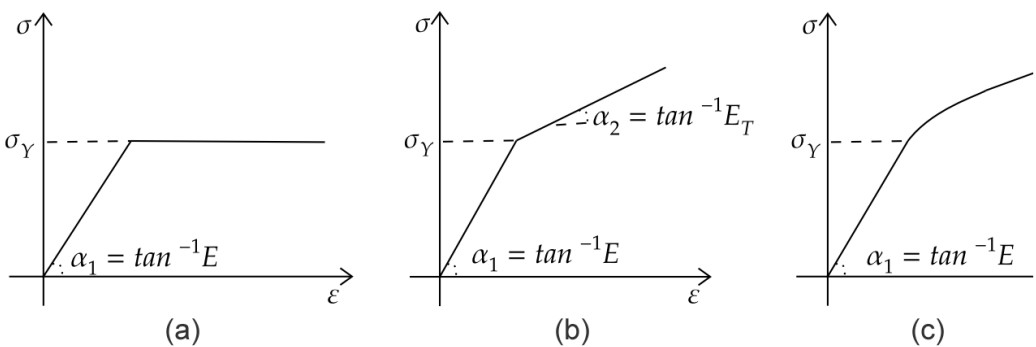

**Figure 4.** Elastoplastic material behavior. (**a**) Elastic-perfectly plastic model. (**b**) Bilinear elastoplastic model. (**c**) Non-linear elastoplastic model.

An isotropic hardening rule is considered. This means that there is a uniform expansion of the initial yield surface with the increase in the effective plastic strain, $\bar{\varepsilon}^p$:

$$\bar{\varepsilon}^p = \int d\bar{\varepsilon}^p = \int \left[ \frac{2}{3} d\bar{\varepsilon}_{ij}^p d\bar{\varepsilon}_{ij}^p \right]^{\frac{1}{2}} \tag{11}$$

in which $d\varepsilon^p$ is the plastic component of the strain occurring during a strain increment.

For the completeness of an elastoplastic model, a flow rule is necessary. It serves as a mathematical description of the evolution of the infinitesimal increments in plastic strain $d\varepsilon^p$ in the course of the load history of the body [29]. The Prandtl–Reuss flow rule is applied:

$$d\varepsilon^p = d\lambda \frac{\partial f}{\partial \sigma} \tag{12}$$

where $d\lambda$ is the plastic multiplier (i.e., proportional constant), and $\frac{\partial f}{\partial \sigma}$ is the $f$ gradient, and is therefore an orthogonal vector to the yield surface [29,30]. Depending on the value of $d\lambda$, the material may be subjected to distinct situations, as shown in Table 1.

**Table 1.** Possible situations resulting from the plastic multiplier and stress point.

| | |
|---|---|
| $d\lambda < 0$ | Elastic unloading occurs (elastic behavior) and the stress point returns inside the yield surface. Hook's law is still valid. |
| $d\lambda = 0$ | Stress point is on the yield surface. Neutral loading (plastic behavior for an elastic–perfectly plastic material). If the material has hardening, this situation corresponds to the beginning of the plasticity. |
| $d\lambda > 0$ | Plastic loading. The stress point is on a constantly expanding yield surface (plastic behavior for a strain-hardening material). |

### 3.1. Elastoplastic Constitutive Matrix

The way to obtain the elastoplastic constitutive matrix, $\mathbf{c}^p$, starts by differentiating Equation (9):

$$dF = \left( \frac{\partial F}{\partial \sigma} \right) d\sigma + \frac{\partial F}{\partial \kappa} d\kappa = 0 \tag{13}$$

Equation (13) can be written as function of $d\lambda$:

$$\mathbf{a}^T d\sigma - A d\lambda = 0 \tag{14}$$

where the vector **a** is the flow vector, $\frac{\partial f}{\partial \sigma}$, and $A$:

$$A = -\frac{1}{d\lambda} \frac{\partial F}{\partial \kappa} d\kappa \tag{15}$$

Using the Prandtl–Reuss flow rule and Hooke's law, the total strain increment, $d\varepsilon$, is obtained as:

$$d\varepsilon = d\varepsilon^e + d\varepsilon^p = \mathbf{c}^{-1} d\sigma + d\lambda \frac{\partial F}{\partial \sigma} \tag{16}$$

where **c** is the elastic constitutive matrix of material properties. Using (14) and (16), one obtains:

$$d\lambda = \frac{1}{[A + \mathbf{a}^T \mathbf{c} \cdot \mathbf{a}]} \mathbf{a}^T d\mathbf{D} d\varepsilon \quad with \quad d\mathbf{D}^T = \mathbf{a}^T \mathbf{D} \tag{17}$$

Substituting Equation (17) in Equation (14) allows one to obtain:

$$d\sigma = \left( \mathbf{c} - \frac{\mathbf{c} \cdot \mathbf{a} \cdot \mathbf{a}^T \mathbf{c}}{\mathbf{a}^T \mathbf{c} \cdot \mathbf{a} + A} \right) d\varepsilon = \mathbf{c}^p d\varepsilon \tag{18}$$

where $\mathbf{c}^p$ is the elastoplastic matrix.

For an isotropic hardening rule, the yield stress is a function of the plastic work,

$$\sigma_Y(\kappa = W_p) \tag{19}$$

where $W_p$ and, consequently, $d\kappa = dW_p$ are:

$$\begin{aligned} \kappa = W_p = \int \sigma_Y d\varepsilon_p = \int \sigma d\varepsilon_p = \int d\lambda \sigma^T \mathbf{a} \\ d\kappa = dW_p = \sigma_Y d\varepsilon_p = \sigma^T d\varepsilon_p = d\lambda \sigma^T \mathbf{a} \end{aligned} \tag{20}$$

Substituting the second equation of (20) into Equation (15), we obtain the hardening constant $A$,

$$A = \sigma^T \mathbf{a} \frac{\partial \sigma_Y}{\partial \kappa} \tag{21}$$

Using the chain rule,

$$\frac{\partial \sigma_Y}{\partial \kappa} = \frac{\partial \sigma_Y}{\partial \varepsilon_p} \frac{\partial \varepsilon_p}{\partial \kappa} \tag{22}$$

and substituting the plastic work rate, $d\kappa$, in Equation (22)

$$\frac{\partial \sigma_Y}{\partial \kappa} = \frac{\partial \sigma_Y}{\partial \varepsilon_p} \frac{1}{\sigma_Y} \tag{23}$$

The partial derivative $\frac{\partial \sigma_Y}{\partial \varepsilon_p}$ is obtained using Young's Modulus, $E$, and the Tangent Modulus, $E_T$ (the slope of the plastic part of the graph in Figure 4b):

$$\frac{\partial \sigma_Y}{\partial \varepsilon_p} = \frac{d\sigma}{d\varepsilon - d\varepsilon_e} = \frac{1}{\frac{d\varepsilon}{d\sigma} - \frac{d\varepsilon_e}{d\sigma}} = \frac{1}{\frac{1}{E_T} - \frac{1}{E}} = \frac{E_T}{1 - \frac{E_T}{E}} = H' \tag{24}$$

and the parameter $A$ is obtained as:

$$A = \frac{H' \sigma^T \mathbf{a}}{\sigma_Y} \tag{25}$$

Euler's theorem shows that for a homogeneous yield function:

$$\frac{\partial f^T}{\partial \sigma}\sigma = \mathbf{a}^T\sigma = \sigma_Y \tag{26}$$

Hence, $A = H'$. The elastoplastic constitutive matrix can finally be written as:

$$\mathbf{c}^p = \left(\mathbf{c} - \frac{\mathbf{c}\cdot\mathbf{a}\cdot\mathbf{a}^T\mathbf{c}}{\mathbf{a}^T\mathbf{c}\cdot\mathbf{a} + H'}\right) \quad \text{with} \quad H' = \frac{E_T}{1 - \frac{E_T}{E}} \tag{27}$$

### 3.2. Non-Linear Solution Algorithm

The use of incremental methods [31] has the intent of achieving an approximate solution to a certain problem with the application of successive load increments, as it is shown in Figure 5a.

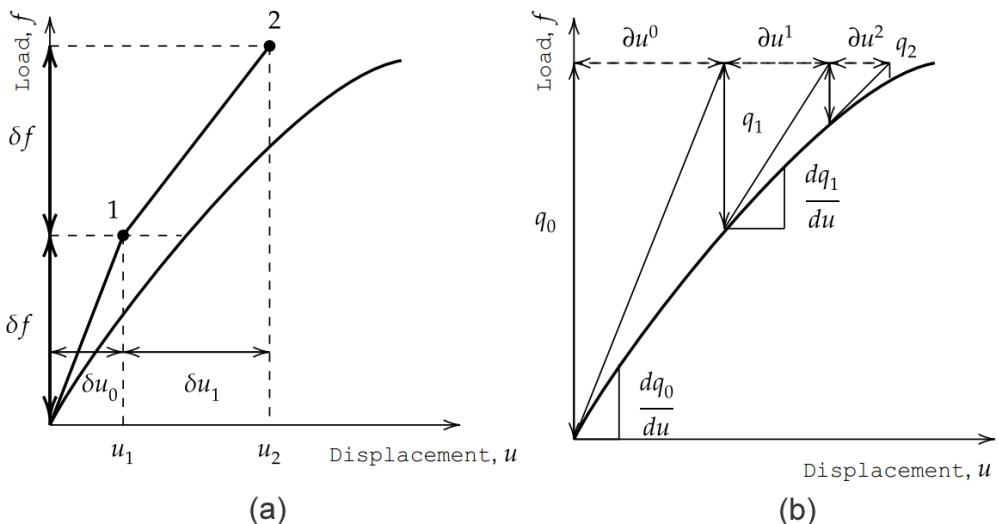

**Figure 5.** Incremental methods. (**a**) Incremental (Euler) scheme. (**b**) Newton–Raphson method.

The displacement, $u$, for each load stage $\delta f$ is given by

$$\delta u = \left(\frac{df}{du}\right)^{-1}\delta f = K_T^{-1}\delta f \tag{28}$$

where $K_T$ is the tangent stiffness.

The biggest issue with the incremental method is the accumulation of errors as the load increments are applied. As it can be seen in Figure 5a, considering point 2 of the diagram, the solution given by the method drifts from the equilibrium curve of the solution. As the load increases, these errors will accumulate. As the nonlinearity of the solution increases, $\frac{df}{du} \to 0$, the number of cumulative errors will increase as well. As a way to overcome this fact, an iterative method is required. One of the most commonly used is the Newton–Raphson incremental scheme. In short, the Newton–Raphson method only provides the displacement, $u$, of a pre-defined load, $f$, as it is shown in Figure 5b.

The iterative changes, $j$, are defined as

$$\partial u^j = \left(\frac{dq^j}{du}\right)^{-1}q^j(u^j) \tag{29}$$

and the displacements are updated using

$$u^j = u^{j-1} + \partial u^j \tag{30}$$

The incremental and iterative methods can be combined (Figure 6), allowing one to obtain the problem solution and a reduction in the lack of equilibrium presented in the incremental process.

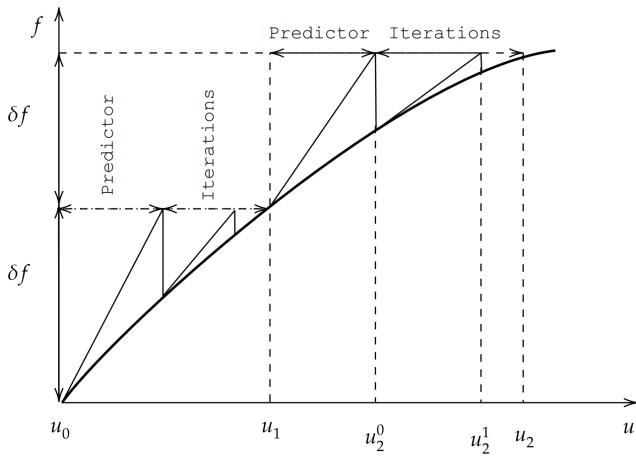

**Figure 6.** Combined incremental and Newton–Raphson method (KT-ALL).

Analyzing Figure 6, the incremental methods act as a predictor agent. The process starts with an initial solution, $u_i^0$—it can be the elastic solution for an arbitrary set of loads—and $u_i^j$ represents the displacement of iteration $j$ for a load increment $i$. The process described in Figure 6 is the full Newton–Raphson method, used to compute the non-linear solutions of the next sections of this paper. In this method, the stiffness matrix of the problem is calculated at the beginning of each increment, and the calculation is repeated for all iterations of said increment. Thus, the stiffness matrix is updated regularly considering the newly updated elastoplastic constitutive matrix. In the same algorithm, a standard backward-Euler scheme is implemented [5] in order to ensure that the stresses that are developed remain inside the yield surface or, at least, very close to it. Hence, the backward-Euler scheme is applied when the stresses of the Gauss points are located outside the yield surface [31]. Thus, the following condition is ensured:

$$f(\sigma) \leq \sigma_Y^* \tag{31}$$

where $f(\sigma)$ is the yield surface, and $\sigma_Y^*$ is the updated yield stress, given by

$$\sigma_Y^* = \sigma + [E_T \cdot \overline{\varepsilon_{i-1}^p}] \tag{32}$$

$\overline{\varepsilon_{i-1}^p}$ is the accumulated effective plastic strain from the previous increment, $(i-1)$. A detailed mathematical explanation of this non-linear algorithm can be found in [32].

## 4. Numerical Results

In this section, 2D and 3D elastoplastic analyses of frame structures are presented. The numerical solutions were obtained through MATLAB algorithms developed in-house combining the presented meshless methods and the elastoplastic model. Whenever possible, the results obtained are compared with solutions documented in the literature.

### 4.1. Elastoplastic Analysis of 2D Frames

In this section, the elastoplastic analysis of selected 2D frames will be presented. The FEM, RPIM, and NNRPIM were used, combined with the full Newton–Raphson non-linear solution algorithm.

#### 4.1.1. Two-Bay Asymmetric Frame

The analyzed frame has its basis in the research paper by Argyris et al. [33]. For this example, a nodal discretization of 6063 nodes was used for the FEM, RPIM, and NNRPIM. The geometry and dimensions of the two-bay asymmetric frame considered are represented in Figure 7. The mechanical properties of the material are given by Table 2.

The base of elements 1 and 5 are completely locked in all degrees of freedom ($u$ and $v$), while a connection at the base of element 4 is considered ($R$), allowing the rotation of the element, with the displacements being locked as well. The length of the elements considered and the cross-section dimensions are shown in Figure 7.

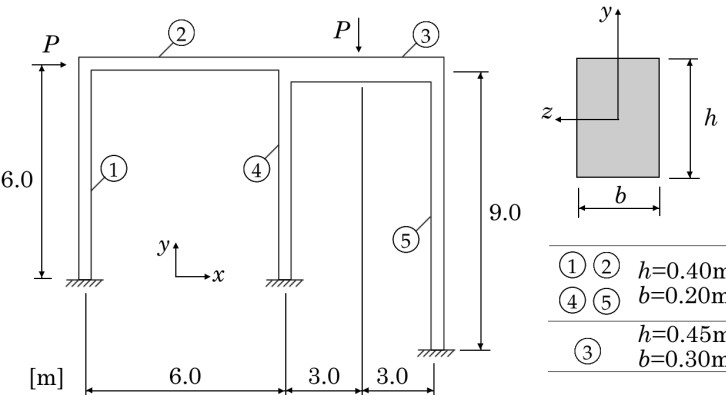

**Figure 7.** Geometry, boundary, and load conditions of the two-bay asymmetric frame.

**Table 2.** Mechanical properties of the two-bay asymmetric frame.

| Mechanical Properties | Value |
|:---:|:---:|
| $E$ [Pa] | $19{,}613.3 \times 10^6$ |
| $E_T$ [Pa] | $19{,}613.3 \times 10^2$ |
| $\nu$ | 0.3 |
| $\sigma_Y$ [Pa] | $98.0665 \times 10^6$ |

The horizontal displacement graphs of point *A*, presented in Figure 7, are now shown in Figure 8. As for units, the *x* axis is normalized by the relationship $\frac{u_A}{6}$, and the *y* axis is correspondent to the load in $\frac{kilopound}{10}$. As it can be seen in Figure 8, the results obtained coincide well with the reference solution for all three numerical methods. As for the marks labeled with respective numbers, these concern the plastic connections developed as the load increases. These plastic connections are represented in Figure 9. Comparing the load values for the six plastic connections calculated using the meshless methods with the FEM solutions, it is possible to obtain the percentage errors shown in Table 3. As expressed in this table, the relative difference between the RPIM results and the FEM is always lower than 6.73%, while in the case of the NNRPIM, the difference is lower than 4.07%. This quantitative comparison highlights the accuracy of the meshless methods studied in this work. The same tendency is verified in the results presented in the coming subsections.

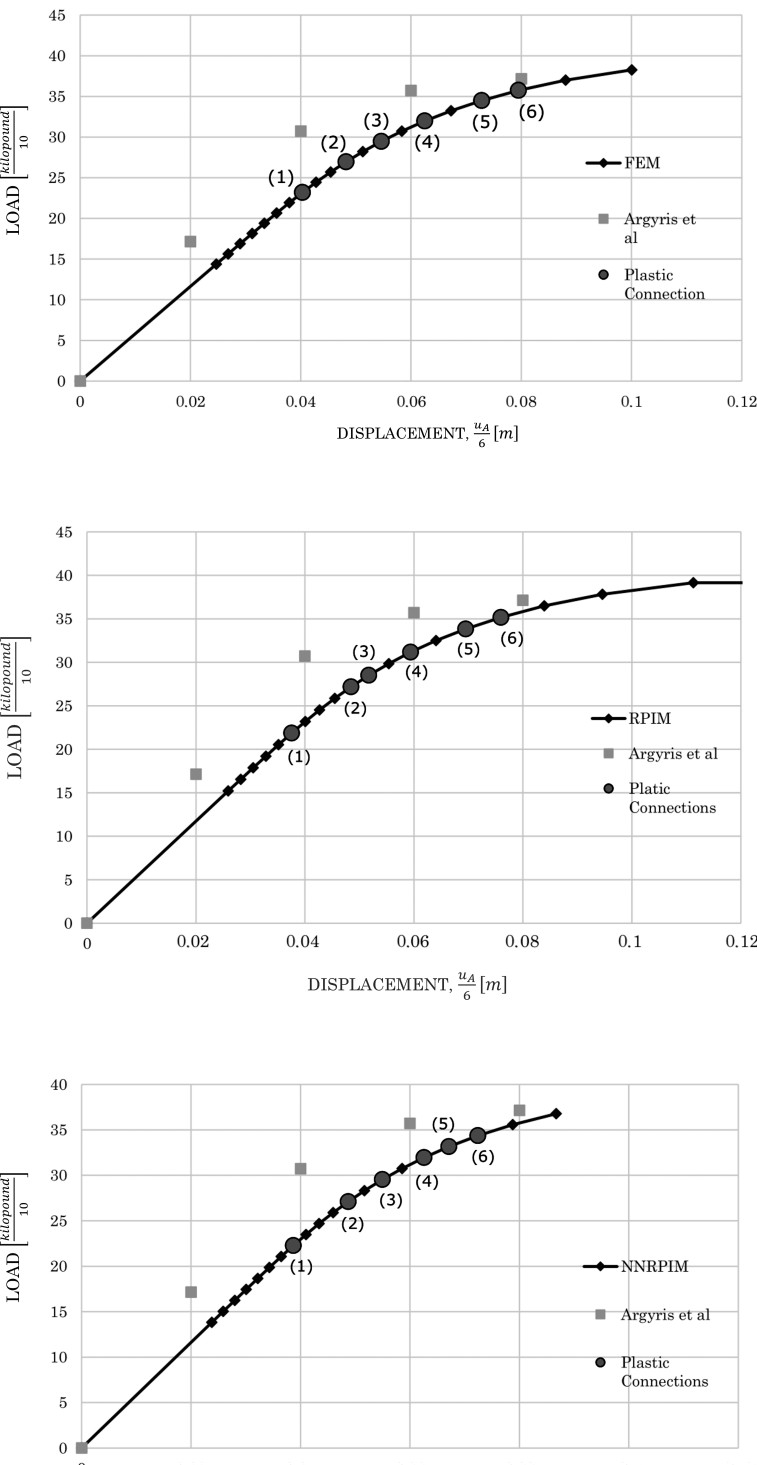

**Figure 8.** Horizontal displacement of point **A** given by the FEM, RPIM, and NNRPIM. The results from Argyris et al. can be found in [33].

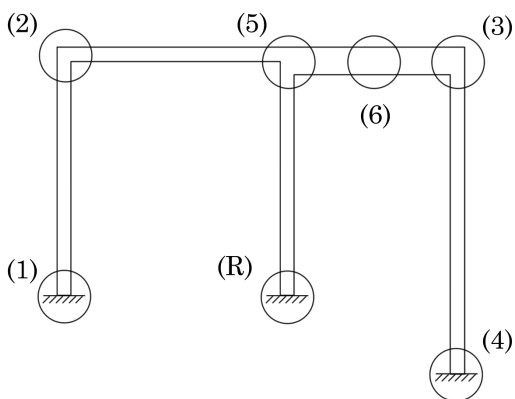

**Figure 9.** Plastic connections in sequential order.

**Table 3.** Load at each plastic connection and percentage difference between the RPIM and NNRPIM solutions with the FEM solution for the 2D case.

| Plastic Connection | Load in $\frac{kilopound}{10}$ | | | % Difference Regarding FEM | |
|:---:|:---:|:---:|:---:|:---:|:---:|
| | FEM | RPIM | NNRPIM | RPIM | NNRPIM |
| 1 | 23.227 | 21.763 | 22.283 | 6.73 | −4.06 |
| 2 | 26.883 | 27.151 | 27.201 | −0.98 | 1.18 |
| 3 | 29.352 | 28.349 | 29.659 | 3.54 | 1.04 |
| 4 | 32.017 | 31.141 | 32.027 | 2.81 | 0.03 |
| 5 | 34.482 | 33.831 | 33.167 | 1.92 | −3.81 |
| 6 | 35.763 | 35.124 | 34.306 | 1.81 | −4.07 |

The effective plastic strain distribution maps for the FEM, RPIM, and NNRPIM are presented in Figures 10–12.

As it can be seen, as the load increases, the amount of effective plastic strain at a certain point of the structure increases, creating plastic connections or plastic hinges. These connections are created due to the plastification of certain points in the structure that are subjected to high stress considering the load input. This way, we can identify the critical points of the structure. Comparing Figures 10–12, the three methods return the same critical points, with the same development of the plastic connections between them.

The diagrams showing the evolution of the effective stress on the identified plastic connections are presented in Figure 13. As can be seen in the figure, for all three numerical methods used, the evolution of the effective stress is quite similar. An interesting detail to be analyzed is the evolution of plastic connections 1 and 2. In fact, plastic connection 2 enters in plastic regime first, when compared to plastic connection 1. Comparing Figures 10–12, in terms of effective plastic strain, both data seem to contradict each other. However, in truth, plastic connection 1 has a very rapid transition between elastic and plastic behavior, while plastic connection 2 is slower in the same transition, as can be assessed by the curvature of both plastic connections shown in the diagrams. Another important aspect of the diagrams presented is the stabilization of the stress level at a certain load applied. After a certain load is applied, the effective stress maintains its value, as it should in a complete plastic regime. We can clearly see the evolution and transition from linear elastic regime to plastic regime, ending with stabilizing the effective stress, even when the load is increased.

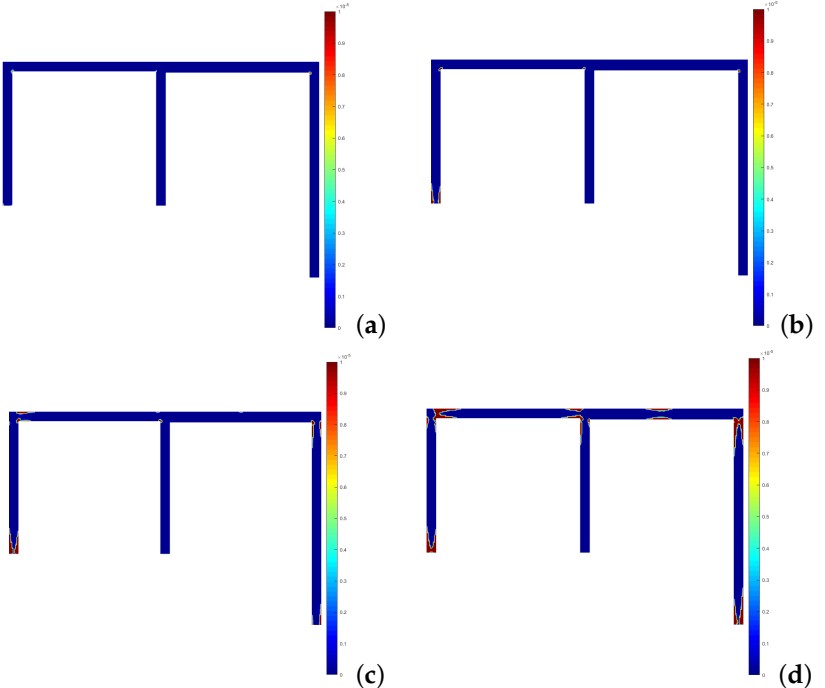

**Figure 10.** Development of the plastic connections for the FEM: (**a**) 19 $\frac{kilopound}{10}$. (**b**) 26 $\frac{kilopound}{10}$. (**c**) 32 $\frac{kilopound}{10}$. (**d**) 36 $\frac{kilopound}{10}$.

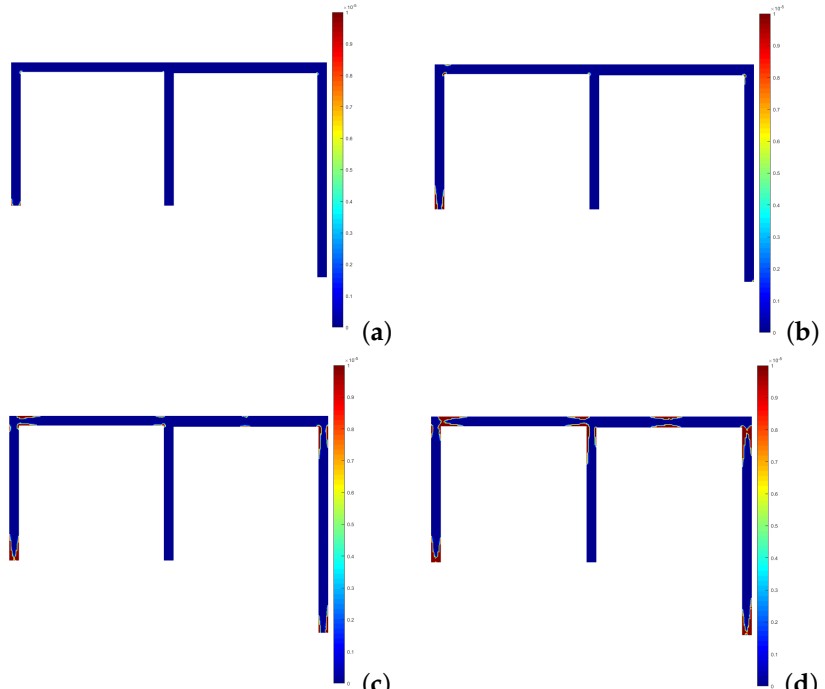

**Figure 11.** Development of the plastic connections for the RPIM: (**a**) 21 $\frac{kilopound}{10}$. (**b**) 27 $\frac{kilopound}{10}$. (**c**) 34 $\frac{kilopound}{10}$. (**d**) 39 $\frac{kilopound}{10}$.

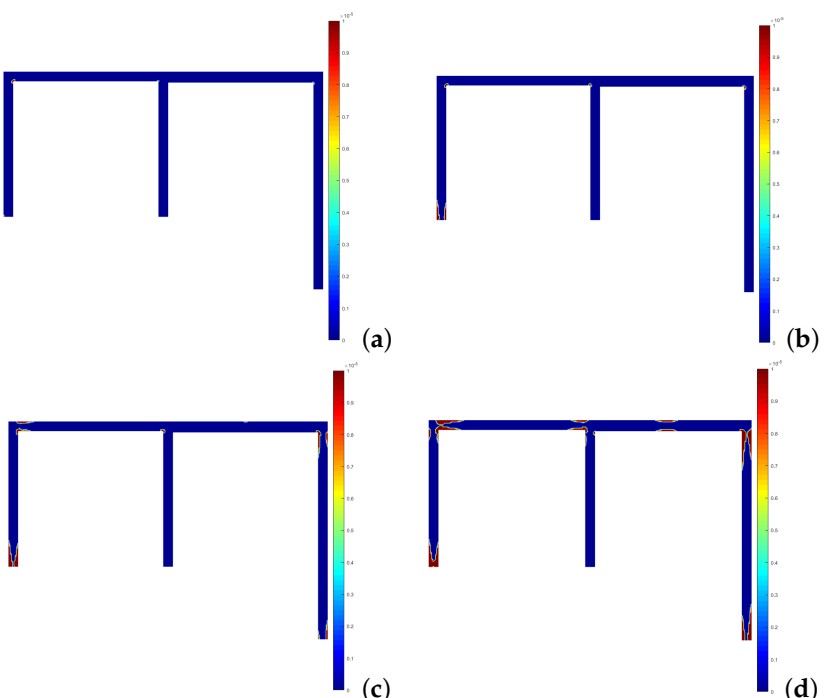

**Figure 12.** Development of the plastic connections for the NNRPIM: (**a**) 19 $\frac{kilopound}{10}$. (**b**) 25 $\frac{kilopound}{10}$. (**c**) 31 $\frac{kilopound}{10}$. (**d**) 37 $\frac{kilopound}{10}$.

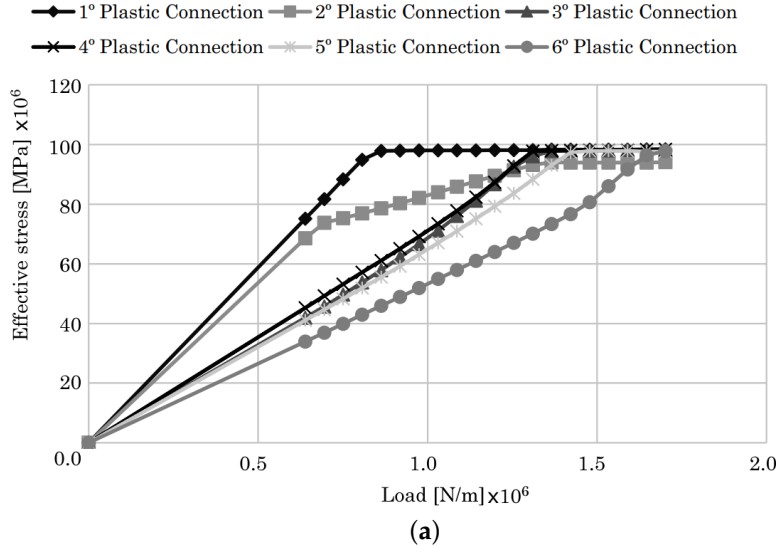

**Figure 13.** *Cont.*

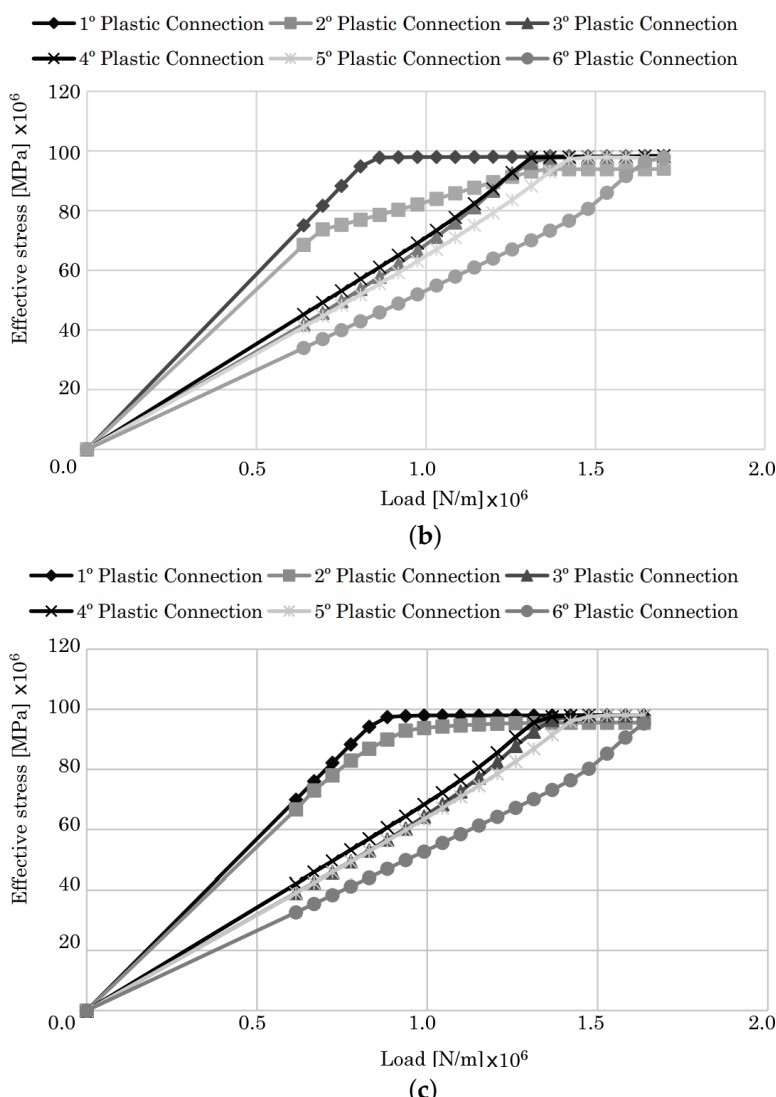

**Figure 13.** Evolution of the effective stress on the plastic connections given by the FEM (**a**), the RPIM (**b**), and the NNRPIM (**c**).

### 4.1.2. Bowstring Bridge

This example was developed by the authors. The purpose of this specific problem is the elastoplastic analysis of a commonly used structure in the industry. Due to the symmetry of the solid considered, only half of the total bridge will be analyzed. The geometry and dimensions of the bowstring bridge considered are represented in Figure 14a.

For the 2D model, the structure was analyzed with a nodal density of 6024 nodes, using the FEM, RPIM, and NNRPIM, combined with the full Newton-Raphson non-linear solution algorithm. Every element present in the structure has a thickness of 100 [mm]. The boundary and loading conditions are represented in Figure 14b. The mechanical properties of the material, common structural steel, are presented in Table 4.

**Table 4.** Mechanical properties of the bowstring bridge.

| Mechanical Properties | Value |
|:---:|:---:|
| $E$ [MPa] | 205,000 |
| $E_T$ [MPa] | 6700 |
| $\nu$ | 0.3 |
| $\sigma_Y$ [MPa] | 230 |

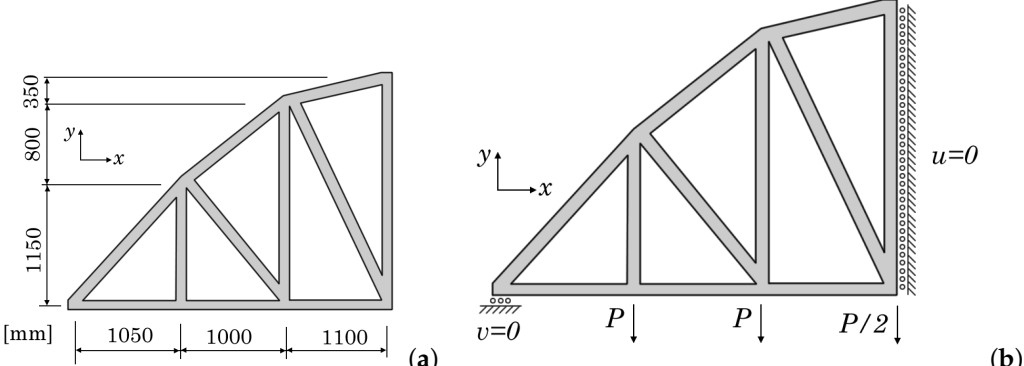

**Figure 14.** Development of the plastic connections for the NNRPIM: (**a**) Dimensions of the bowstring bridge in mm. (**b**) Boundary and load conditions of the bowstring bridge.

The vertical displacement of point **A**, represented in Figure 14, for all three numerical methods, is presented in Figure 15.

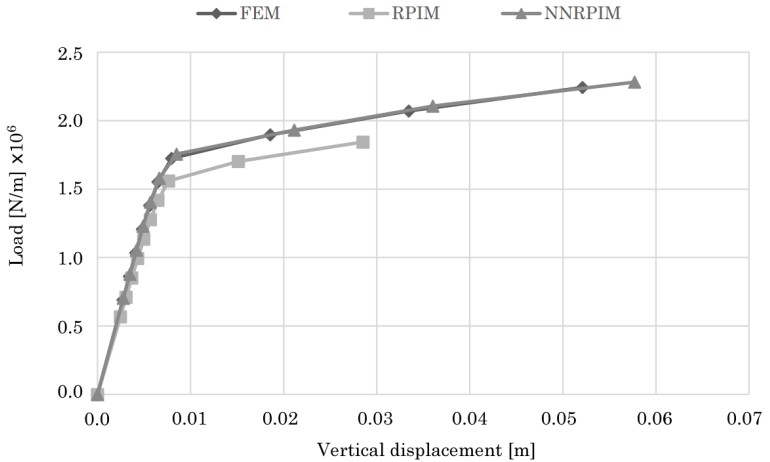

**Figure 15.** Vertical displacement of point **A** of the 2D bowstring bridge.

As it can be seen in Figure 15, the FEM and NNRPIM obtain almost equal results, in that the load/displacement curves for the two numerical methods are almost equal to each other. As for the RPIM curve, it follows the tendency of the FEM and NNRPIM up to a certain point. It then deviates to a lower level when compared with the other two curves. Nevertheless, all three results are very similar to each other, and good agreement between them exists.

The effective plastic strain distribution maps for the FEM, RPIM, and NNRPIM, are presented in Figures 16–18.

Analyzing Figures 16–18, we can clearly see the critical areas of the structure as the load increases. For all three methods, the plastic strain begins at the same point and propagates extensively along the upper arch of the structure. Said arch is under compressive load, while the lower beam is under traction load. As plastification ensues, the load applied to the bridge is mainly supported by this lower beam. The distribution of compressive and traction stress is due to the trusses included in the design of the structure.

As for the comparison between the three numerical methods, the FEM and NNRPIM return almost equal results, which is to be expected given the almost equal load/displacement diagram shown in Figure 15. As for the RPIM, the plastification of the structure is not as broad as in the other two numerical methods, due to the fact that the load applied is not as great as in the FEM and NNRPIM, which limits the plastic strain.

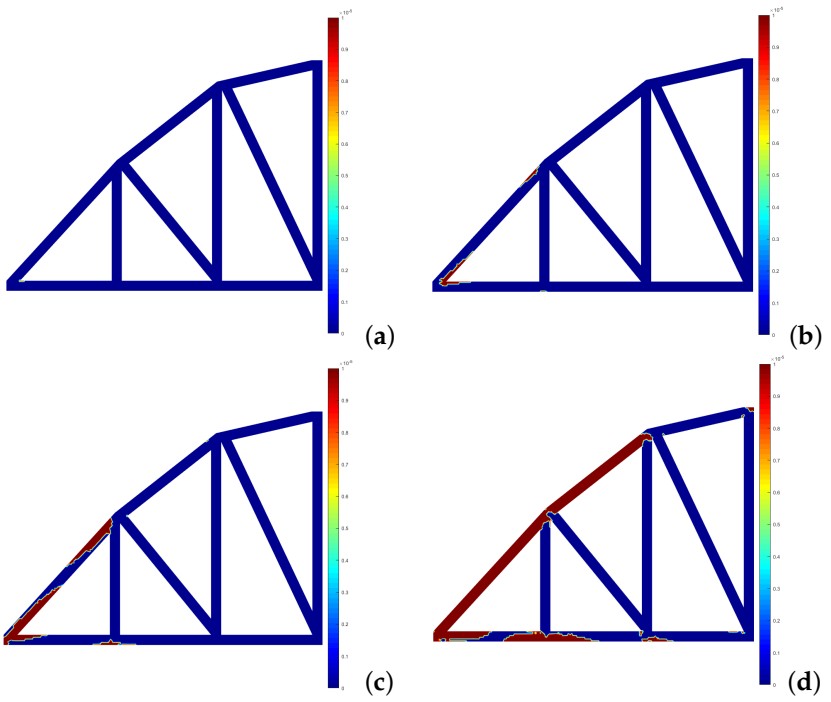

**Figure 16.** Development of the effective plastic strain for the FEM: (**a**) $8.62 \times 10^5$ N/m. (**b**) $1.38 \times 10^6$ N/m. (**c**) $1.72 \times 10^6$ N/m. (**d**) $2.24 \times 10^6$ N/m.

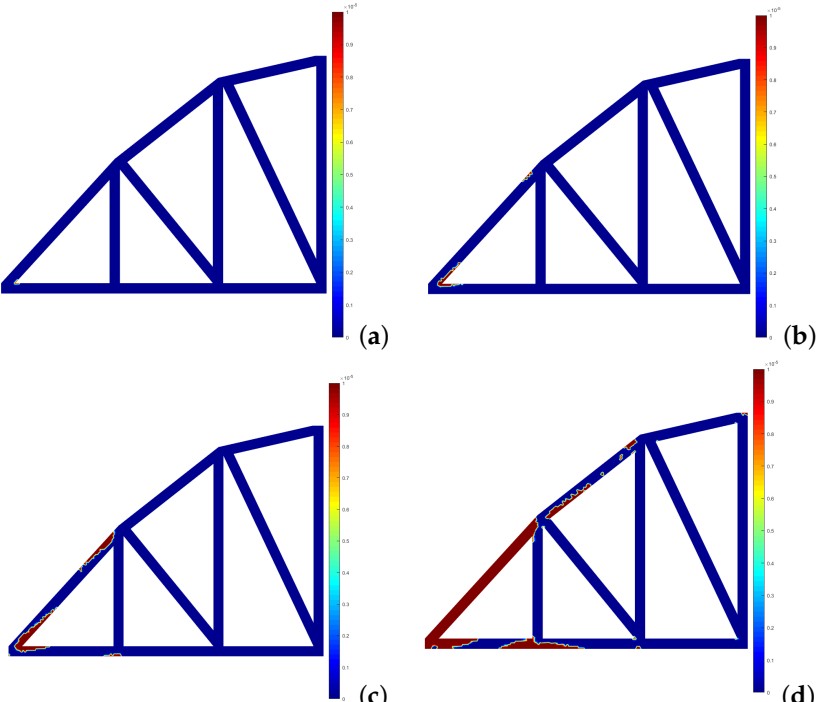

**Figure 17.** Development of the plastic connections for the RPIM: (**a**) $7.09 \times 10^5$ N/m. (**b**) $1.13 \times 10^6$ N/m. (**c**) $1.42 \times 10^6$ N/m. (**d**) $2.24 \times 10^6$ N/m.

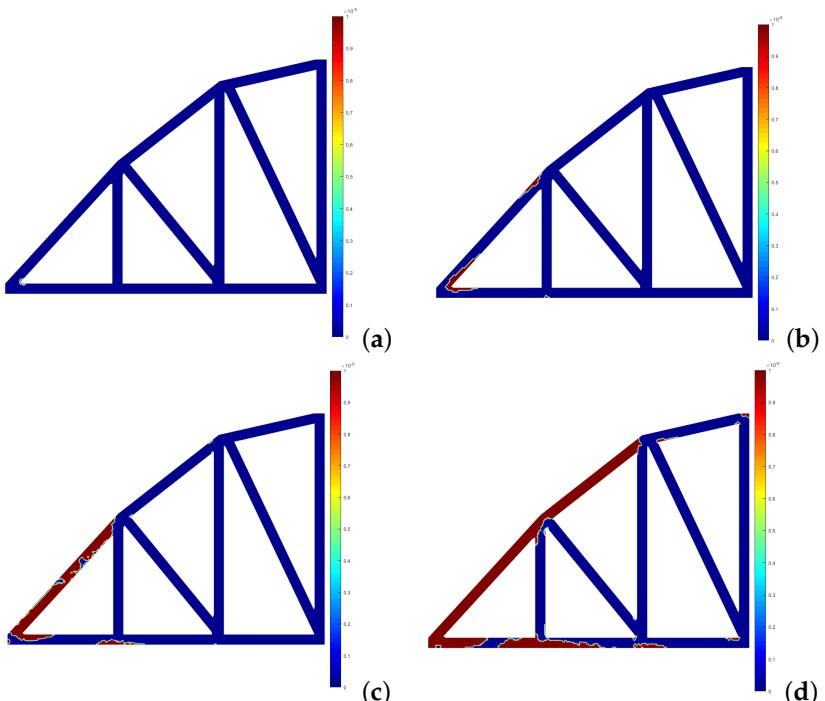

**Figure 18.** Development of the plastic connections for the NNRPIM: (**a**) $8.77 \times 10^5$ N/m. (**b**) $1.40 \times 10^6$ N/m. (**c**) $1.75 \times 10^6$ N/m. (**d**) $2.28 \times 10^6$ N/m.

*4.2. Elastoplastic Analysis of 3D Structures*

In this section, the analysis of the 2D structures presented earlier will be made, but considering 3D geometry.

4.2.1. Two-Bay Asymmetric Frame

The same dimensions used in the 2D case are applied here as well. The thickness variation for the 3D model is represented in Figure 19, as an upper view of the model. Only half of the structure is analyzed due to symmetry. As for nodal density, a discretization of 2481 nodes was used.

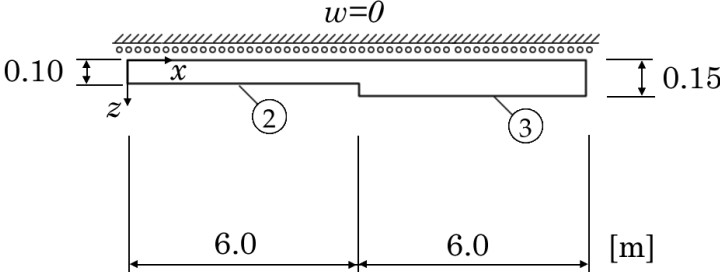

**Figure 19.** Thickness variation and additional boundary condition for the 3D model—Upper view.

Additionally, due to symmetry, the $w$ displacement is locked, which is also represented in Figure 19. Similarly with the 2D case, for the 3D case, the horizontal displacement of point **A** is presented in Figure 20, and it is compared with the reference solution. The units used are the same as in the 2D case. As it can be seen in the referred figure, although not as coincident with the reference solution given by Argyris et al. when compared with the 2D solutions, the obtained results are nonetheless similar to the reference values. This deviation is mainly caused by the nodal density used. For 3D geometries, the computational cost of a high nodal density, considering the computational power that is available, was very high. Therefore, considering the time that it would take for the calculations to be complete, the nodal density used was considered satisfactory.

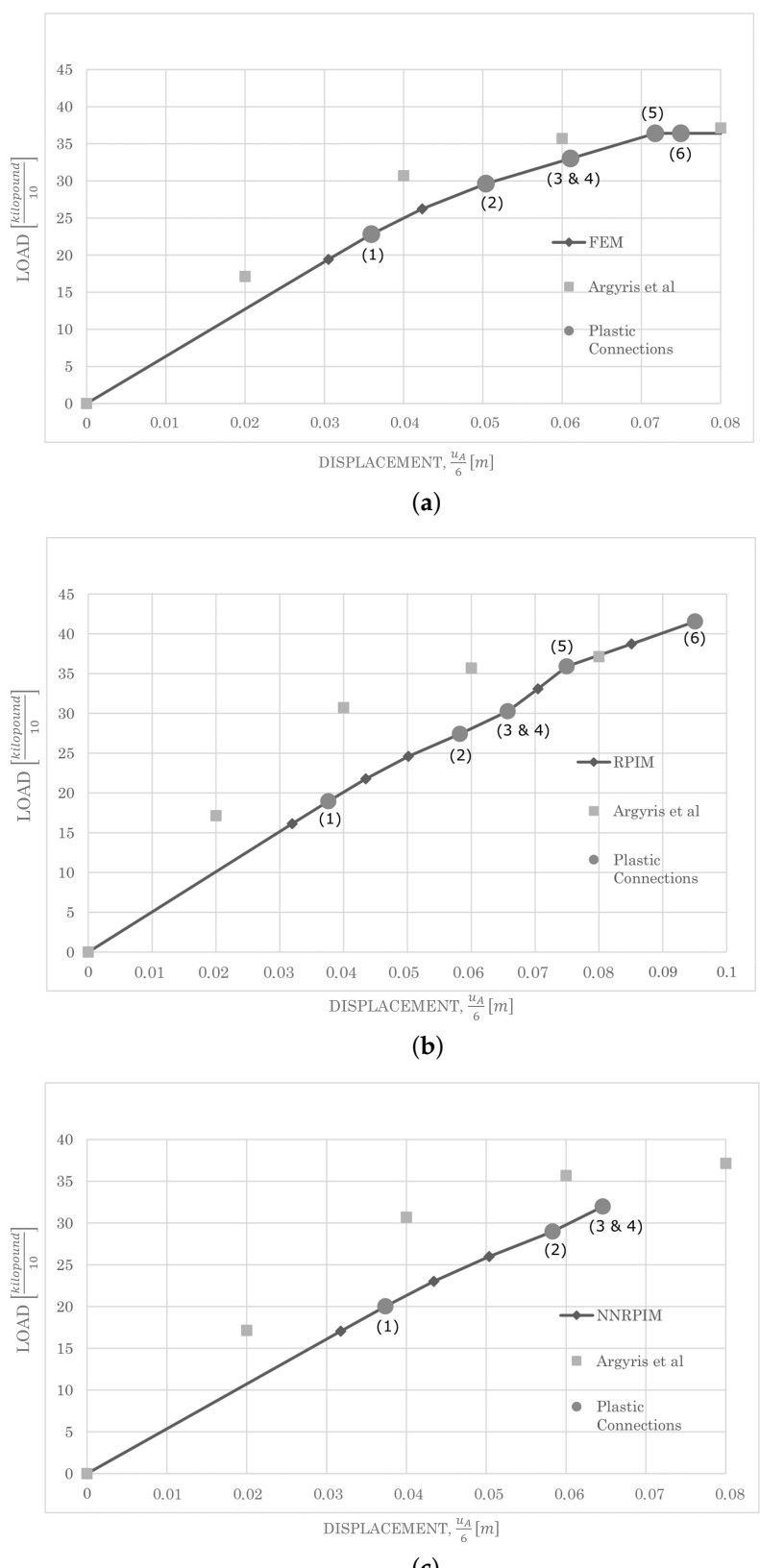

**Figure 20.** Horizontal displacement of point **A** given by the FEM (**a**), the RPIM (**b**), and the NNRPIM (**c**). The results from Argyris et al. can be found in [33].

The effective plastic strain distribution maps for the FEM, RPIM, and NNRPIM, considering the 3D case, are presented in Figures 21–23.

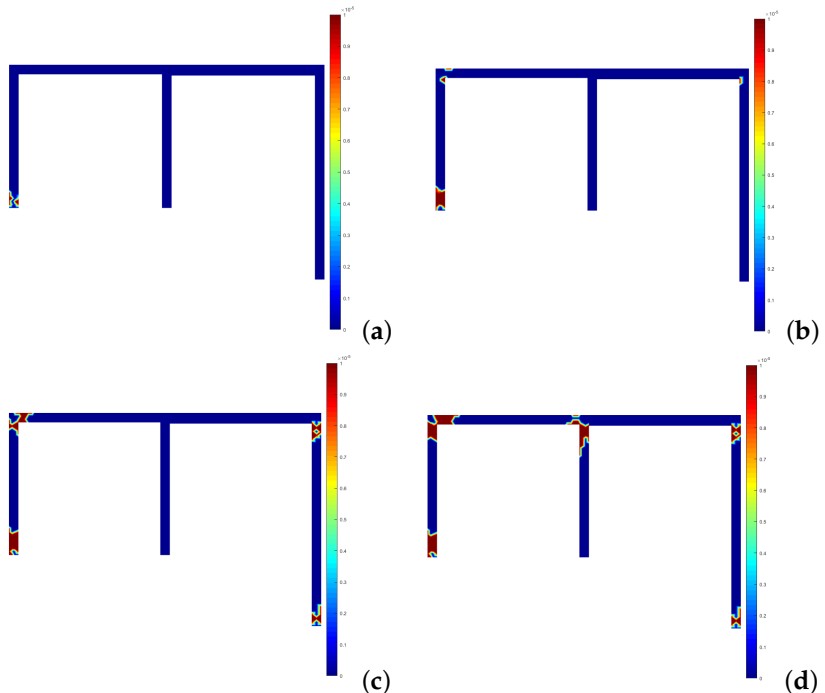

**Figure 21.** Development of the plastic connections for the FEM—3D case: (**a**) 23 $\frac{kilopound}{10}$. (**b**) 30 $\frac{kilopound}{10}$. (**c**) 33 $\frac{kilopound}{10}$. (**d**) 36 $\frac{kilopound}{10}$.

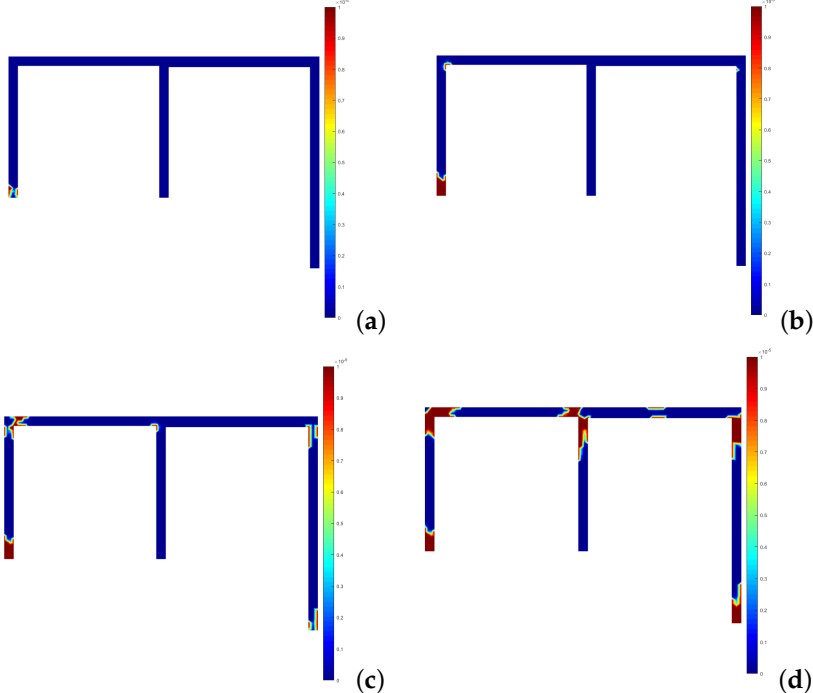

**Figure 22.** Development of the plastic connections for the RPIM—3D case: (**a**) 19 $\frac{kilopound}{10}$. (**b**) 25 $\frac{kilopound}{10}$. (**c**) 30 $\frac{kilopound}{10}$. (**d**) 42 $\frac{kilopound}{10}$.

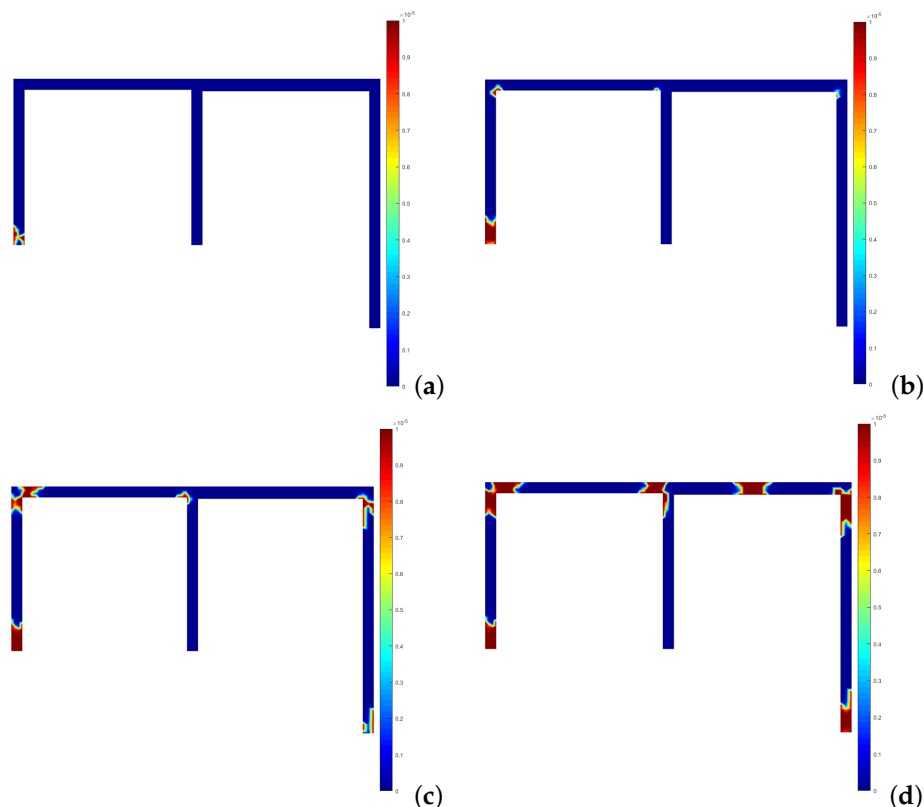

**Figure 23.** Development of the plastic connections for the NNRPIM—3D case: (**a**) $20 \frac{kilopound}{10}$. (**b**) $26 \frac{kilopound}{10}$. (**c**) $32 \frac{kilopound}{10}$. (**d**) $44 \frac{kilopound}{10}$.

Analyzing Figures 21–23 and comparing them to the 2D case, the same plastic connections are developed. In the 3D case, these plastic connections are more broad, meaning that the plastic strain on these critical points is more severe.

In terms of the progression of the plastic connections, as the load increases, for all three numerical methods, we can see the same evolution. An interesting detail to point out is the absence of plastic connection 6 in Figure 21d. The plastic connection is not shown due to the fact that the load increment after $36 \frac{kilopound}{10}$ leads to the complete hardening of the material, and due to that fact, the plastic connection is not evident. However, as it is shown in Figure 20a, this plastic connection is developed but at a lower load. Overall, the results are in agreement, as they should be, with the 2D case. The diagrams showing the evolution of the effective stress on the identified plastic connections are presented in Figure 24. The results obtained by the three methods are similar. The same detail presented for the 2D case, concerning the entrance of plastic connection 2 in the elastoplastic regime at first, with plastic connection 1 having a rapid transition between elastic and plastic behavior, is also evident. The simultaneous entrance into plasticity of plastic connections 3 and 4 is also evident since both curves shift at the same time.

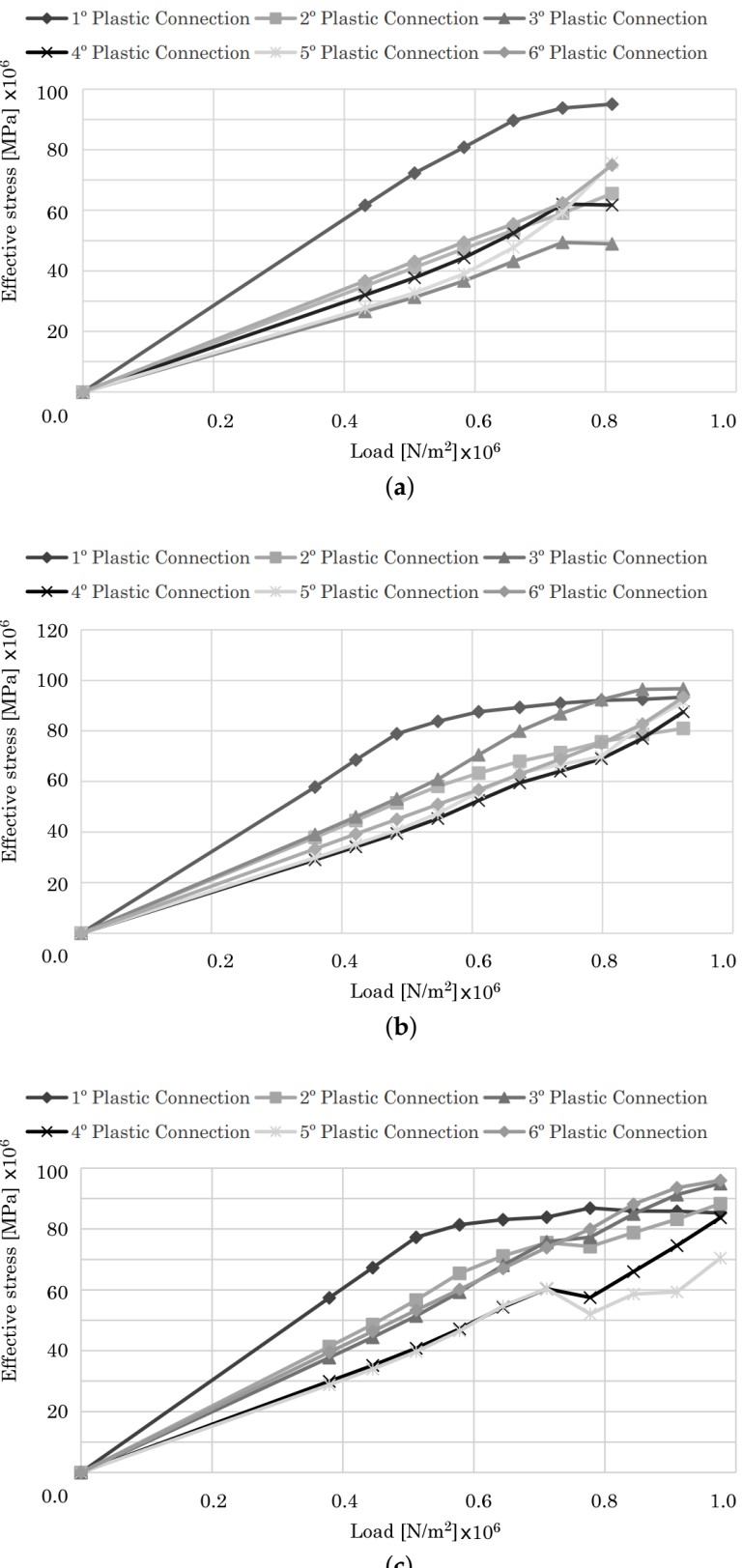

**Figure 24.** Evolution of the effective stress on the plastic connections given by the FEM (**a**), the RPIM (**b**), and the NNRPIM (**c**)—3D case.

The results presented for the horizontal displacement of point **A** for the 2D and 3D case will now be presented in a combined diagram, in Figure 25, for a more complete comparison.

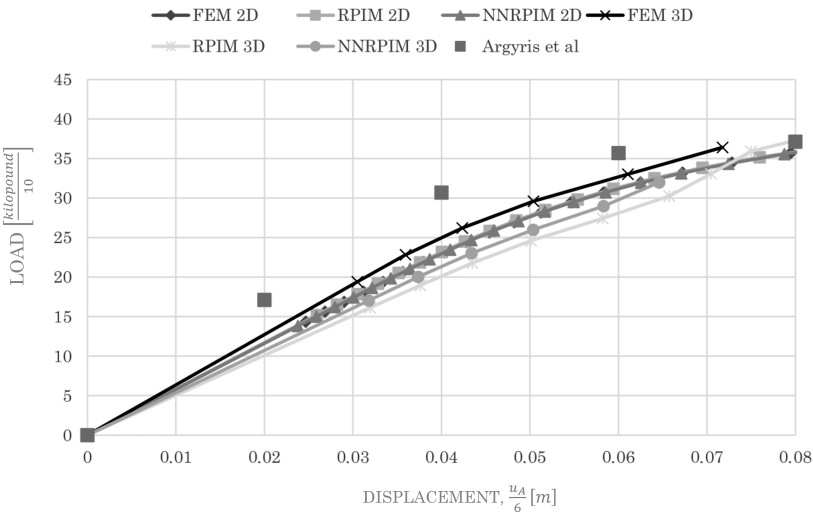

**Figure 25.** The 2D and 3D results for the horizontal displacement of point **A**. The results are compared with the solution of Argyris et al. [33].

The combined results shown in Figure 25 show good agreement between the 2D and 3D results, for all three numerical methods, and a combined agreement with the reference solution by Argyris et al. [33].

### 4.2.2. Bowstring Bridge

As was carried out with the two-bay asymmetric frame, the 3D case of the bowstring bridge will be analyzed. The 3D geometry consists of a complete extrusion of the 2D section with a thickness of 0.1 (m). A nodal density of 3032 nodes was used. The mechanical properties and dimensions are the same.

The vertical displacement of point **A**, for all three numerical methods, is presented in Figure 26.

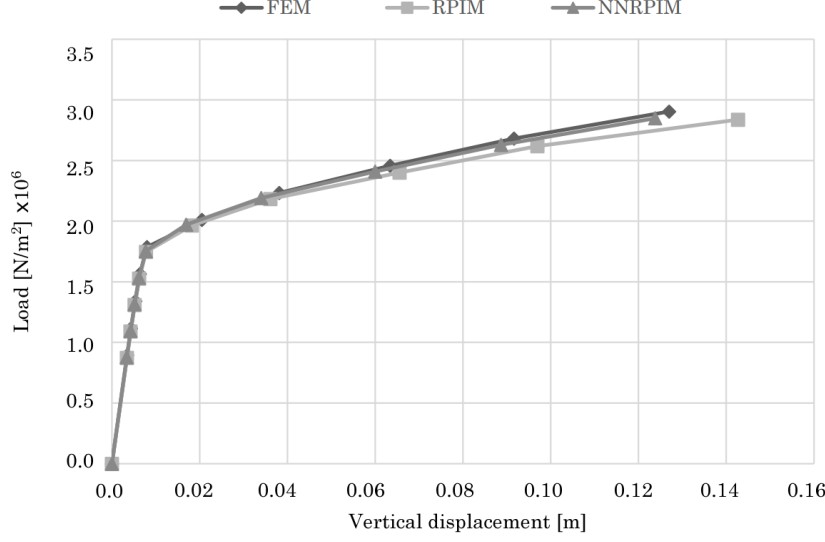

**Figure 26.** Vertical displacement of point *A* of the 3D bowstring bridge.

The results presented in Figure 26 show excellent agreement between the three numerical methods used, and are also in agreement with the values obtained for the 2D case presented earlier.

The development of the effective plastic strain along the structure as the load increases is presented in Figures 27–29.

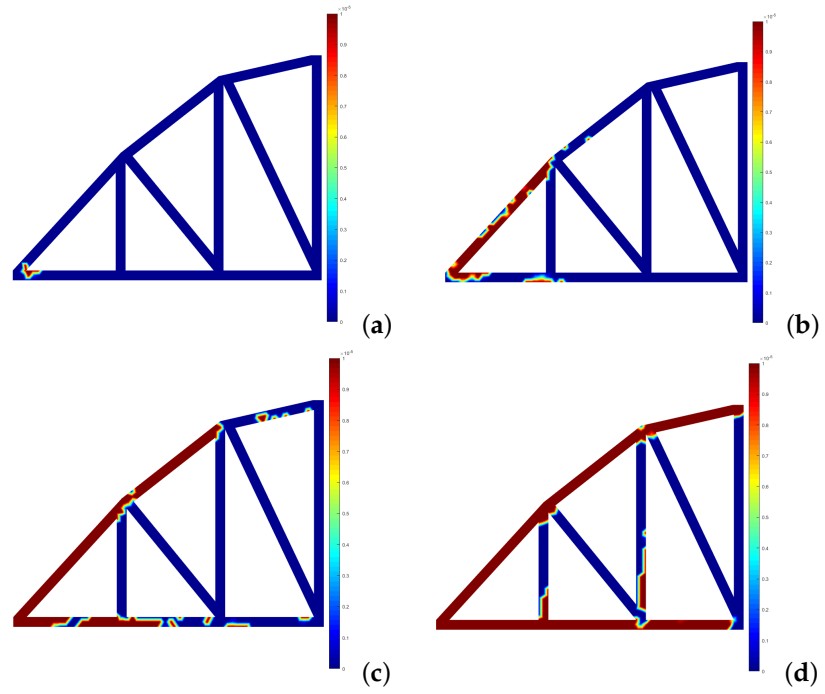

**Figure 27.** Development of the effective plastic strain for the FEM—3D case: (**a**) $1.12 \times 10^6$ N/m. (**b**) $1.79 \times 10^6$ N/m. (**c**) $2.23 \times 10^6$ N/m. (**d**) $2.90 \times 10^6$ N/m.

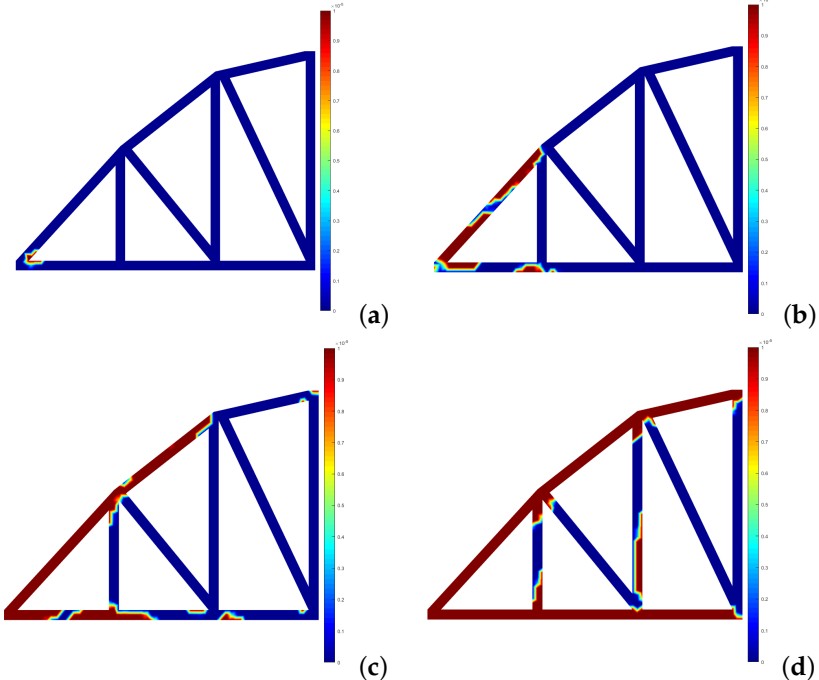

**Figure 28.** Development of the effective plastic strain for the RPIM—3D case: (**a**) $1.09 \times 10^6$ N/m. (**b**) $1.75 \times 10^6$ N/m. (**c**) $2.18 \times 10^6$ N/m. (**d**) $2.84 \times 10^6$ N/m.

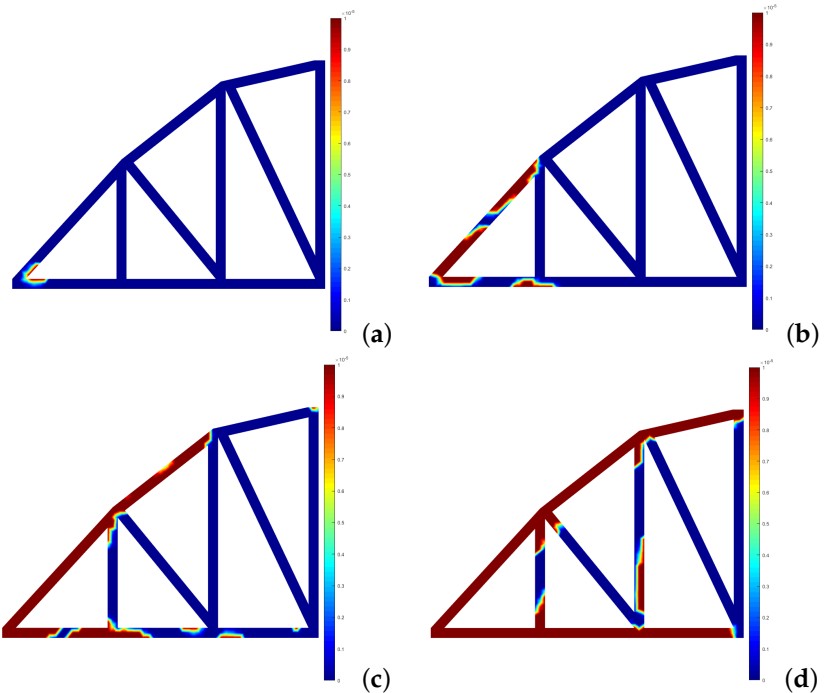

**Figure 29.** Development of the effective plastic strain for the NNRPIM—3D case: (**a**) $8.77 \times 10^5$ N/m. (**b**) $1.40 \times 10^6$ N/m. (**c**) $1.75 \times 10^6$ N/m. (**d**) $2.28 \times 10^6$ N/m.

The results obtained are in good agreement with the 2D results. The plastic strain initiates on the same point and propagates through the same areas. In the 3D case, the plastic strain is higher and affects the support trusses of the bridge, while in the 2D case, the plastic strain barely reaches these elements. Analyzing the three numerical methods, the results are very similar between them.

The combined results for the vertical displacement of point **A** are presented in Figure 30.

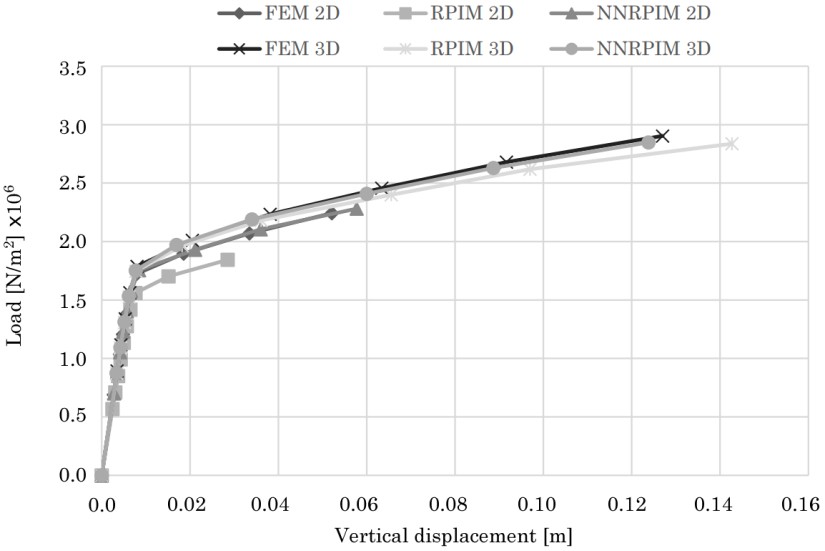

**Figure 30.** The 2D and 3D results for the vertical displacement of point *A*.

The results shown in Figure 30 show excellent agreement between the three numerical methods for both cases. The 3D case returns higher vertical displacements due to a higher load input. However, for the same loads, the vertical displacement is very similar.

## 5. Conclusions

In this work, the Radial Point Interpolation Method (RPIM) and the Natural Neighbor Radial Point Interpolation Method (NNRPIM) were used in the elastoplastic analysis of 2D and 3D structural elements.

A selected example from Argyris et al. and a novel example by the authors are presented. It is concluded that for the two-bay asymmetric frame, considering both 2D and 3D analyses, the FEM, the RPIM, and the NNRPIM reach a good agreement with the reference solution. The three methods reach the same results not only in terms of order but also in terms of the location of the plastic connections. For the 3D case, a higher nodal density for a more accurate analysis is needed, with this fact being dependent on the computational power that is available. For the bowstring bridge, excellent agreement for all three methods is reached, with the RPIM for the 2D case not reaching the load input to match the other methods. In both 2D and 3D examples, the development of the plastic hinges and overall material plasticity is broader in the 3D case, mainly due to the lower nodal density used. The numerical studies developed throughout this work show that the RPIM and NNRPIM reach good results not only when compared to the FEM but with the reference solutions presented, showing their accuracy and capabilities to analyze the same kind of engineering problems analyzed with the FEM until the present day and even more problems for which the FEM may not be the most suitable tool—such as a large deformations formulation. It was shown that these meshless methods behave better than the FEM for low nodal densities, with the NNRPIM having a disadvantage in terms of preprocessing time. The RPIM, depending on the Gauss quadrature chosen, has a low preprocessing time phase.

**Author Contributions:** J.B. Conceptualization; J.B., methodology; J.B. and D.E.S.R., software; M.A. and J.B., validation; M.A. and J.B., formal analysis; J.B., investigation, J.B., resources; J.B. and D.E.S.R., data curation; M.A. and D.E.S.R., writing—original draft preparation; J.B. and D.E.S.R., writing—review and editing; J.B. and D.E.S.R., visualization; J.B. and D.E.S.R., supervision; J.B., project administration; J.B., funding acquisition. All authors have read and agreed to the published version of the manuscript.

**Funding:** The authors acknowledge the funding provided by Ministério da Ciência, Tecnologia e Ensino Superior—Fundação para a Ciência e a Tecnologia (Portugal) and LAETA, under internal project UIDB/50022/2020.

**Data Availability Statement:** Data are contained within the article.

**Conflicts of Interest:** The authors declare no conflict of interest.

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
