# Peer review of "Elastoplastic Analysis of Frame Structures Using Radial Point Interpolation Meshless Methods"

_applsci, doi:10.3390/app132312591_

Round 1
Reviewer 1 Report
Comments and Suggestions for Authors
This article is well defined and acceptable in its present form. Only 1 correction that Out of 27 citations 07 belongs to Authors which is very high.
Author Response
Dear Reviewer #1,
The point-by-point response can be found in the attached document.
Best regards,

Reviewer 2 Report
Comments and Suggestions for Authors
Dear Authors,
beyond some minor corrections highlighted in the attached file, there are some major defects:
- it is really hard to understand the overall scope of the paper.
- From page 1 to 9, paragraph 1 to 3, a theory is detailed, with lots formula and considerations and pictures. But it seems this theory was not developed inside your research, just applied. In this case, such sections have to be deleted in the way to focus the paper on your own research. Otherwise, since there is not clear evidence about the connection between theory and your application, the residual paragraphs have to be changed for the scope.
- methods & tools are not fully understandable (submerged inside other information). It is better to define a specific paragraph.
- innovative aspects are not properly introduced and detailed.

Author Response
Dear Reviewer #2,
The point-by-point response can be found in the attached document.
Best regards,

Reviewer 3 Report
Comments and Suggestions for Authors
Review of the manuscript:
Elastoplastic Analysis of Frame Structures using Radial Point Interpolation Meshless Methods
1.The findings are sufficiently novel to warrant publication.
2. The conclusions are adequately supported by the data presented.
3. The article is clearly and logically written so that it can be understood by one who is not an expert in the specific field. The work provides an important contribution to its field, consistent with the scope of the journal. The better discussion is needed.
The paper is describing the actual problematics. Authors deals with an advanced discretization techniques (i.e. meshless methods), which are applied in the elastoplastic analysis of 2D and 3D structural elements. The Radial Point Interpolation Method (RPIM) and the Natural Neighbour Radial Point Interpolation Method (NNRPIM) are combined with a non-linear iterative algorithm, fully developed by the authors, with the objective of analyzing for the first time the elastoplastic behavior of a two-bay asymmetric frame and bowstring bridge considering 2D and 3D analysis.
Comments:
The description and discussion of the uncertainties of the calculations is needed in the paper.
Only qualitative and no quantitative description of the results was realized in the paper.
The better discussion is needed.
Figures 8,9 and 10: Please mark and describe the axis of the graphs (quantity and unit)
Please describe the calculations of the uncertainties of the horizontal displacement of point A on the Figures 8,9 and 10. Please introduce the values or the graphs of the uncertainties.
Please describe the calculations of the uncertainties of the evolution of the effective stress on the plastic connections on the Figures 15,16 and 17. Please introduce the values or the graphs of the uncertainties.
Please describe the calculations of the uncertainties of the Load and vertical displacements on the Figures 19,16 and 17. Please introduce the values or the graphs of the uncertainties.
Please describe the calculations of the uncertainties of the horizontal displacement of point A on the Figures 24,25 and 26. Please introduce the values or the graphs of the uncertainties.
Please describe the calculations of the uncertainties of the evolution of the effective stress on the plastic connection on the Figures 30,31 and 32. Please introduce the values or the graphs of the uncertainties.
Please describe the calculations of the uncertainties of the 2D and 3D results for the horizontal displacement of point A on the Figure 33. Please introduce the values or the graphs of the uncertainties.
Please describe the calculations of the uncertainties of the Vertical displacement of point A of the 3D bowstring bridge on the Figure 34. Please introduce the values or the graphs of the uncertainties.
Please describe the calculations of the uncertainties of the 2D and 3D results for the vertical displacement of point A on the Figure 38. Please introduce the values or the graphs of the uncertainties.

Author Response
Dear Reviewer #3,
The point-by-point response can be found in the attached document.
Best regards,

Reviewer 4 Report
Comments and Suggestions for Authors
In this paper, meshless methods are applied in the elastoplastic analysis of 2D and 3D structural elements, capable of producing more accurate and smoother strain and stress fields. The method developed in the paper has been compared with traditional FE calculation method. The work done in this paper is impressive, which is meaningful in this field. However, the writing of the manuscript needs to be improved. The paper is recommended to be published in Applied Sciences after answering following questions and finishing corresponding revision in the manuscript.
(1) The existing work about elastoplastic/nonlinear analysis using meshless method in open literature should be cited and introduced in the Introduction section. The novelty and main work done in this work compared with the existing ones should be highlighted then. The advantages of RPIM and NNRPIM should be clarified.
(2) Figures like Fig. 9, Fig. 10, Fig. 16, Fig. 25, Fig. 30, Fig. 31, etc. should be plotted using different colors to make the comparison between different curves more noticeable.
(3) There are too many figure results in the manuscript. Some similar figures should be arranged as 2x2 subplots with shared legend to reduce the space they occupied in the manuscript. Or could be different curves plotted in same figure? Otherwise, some figures could be put in Appendix.
(4) In page 10, there is large blank space within the text. This should be avoided.
(5) One of the purposes of Fig. 12-14 and Fig. 20-22 is to compare the field plot calculated by different method. So it’s better to put the three plots with same load but calculated by different method in one figure. This will make it easier for reader to compare different methods.
Comments on the Quality of English Languagenone
Author Response
Dear Reviewer #4,
The point-by-point response can be found in the attached document.
Best regards,

Round 2
Reviewer 2 Report
Comments and Suggestions for Authors
No further changes needed.